Malagasy Polka Dot Moths (Noctuoidea: Erebidae: Arctiinae: Syntomini) of Ambohitantely—endemism in the most important relict of Central Plateau rainforest in Madagascar

http://orcid.org/0000-0002-4885-1301 Wiorek Marcin 1
Malik Kamila 2
Lees David 3
Przybyłowicz Łukasz 1 lukasz@isez.pan.krakow.pl
1 Institute of Systematics and Evolution of Animals, Polish Academy of Sciences , Kraków , Poland
2 Department of Invertebrate Evolution, Institute of Zoology and Biomedical Research, Jagiellonian University , Kraków , Poland
3 Department of Life Sciences, Natural History Museum , London , United Kingdom
Bond Jason
Electronic publication date: 2021 Jun 24
Publication date: 2021
Volume: 9
Electronic Location ID: e11688
Received 2021 Mar 30; Accepted 2021 Jun 7
Copyright: © 2021 Wiorek et al.
Copyright year: 2021
Copyright holder: Wiorek et al.
License: This is an open access article distributed under the terms of the Creative Commons Attribution License, which permits unrestricted use, distribution, reproduction and adaptation in any medium and for any purpose provided that it is properly attributed. For attribution, the original author(s), title, publication source (PeerJ) and either DOI or URL of the article must be cited.
License URL: https://creativecommons.org/licenses/by/4.0/

Keywords: Madagascar, Central Plateau, Syntomini, Endemism, Evolutionary radiation, Intraspecific variation, Taxonomy

Funding: National Science Centre, Poland 2018/29/B/NZ8/00186 Our research was funded by National Science Centre, Poland (Grant No. 2018/29/B/NZ8/00186). The funders had no role in study design, data collection and analysis, decision to publish, or preparation of the manuscript.

==============================
Malagasy Syntomini (Polka Dot Moths) are one of the largest endemic lineages of Lepidoptera on the island, belonging to the Tiger Moth subfamily (Arctiinae). This diverse radiation comprises nearly 100 valid described species that share a single ancestor. Despite a monograph in 1964 by Paul Griveaud, systematics of the group greatly needs modern revision, and their distribution on the island is still poorly known. This contribution concerns the diversity of Syntomini of the Réserve Spéciale d’Ambohitantely, which protects the largest remaining, but already highly fragmented, vestige of Central Plateau rainforest in Madagascar. Here we provide an annotated checklist of the eight species occurring in the Reserve. Two species are recorded from the forest for the first time, while five endemics are until now known only from Ambohitantely. We also describe for the first time the female of Thyrosticta vestigii Griveaud, 1964 and of Maculonaclia tampoketsya Griveaud, 1969, as well as a yellow morphotype of Thyrosticta dilata Griveaud, 1964, and we redescribe and illustrate the genitalia of the remaining species. The significance of such colour pattern variation in aposematic moths and the role of this Reserve as a local centre of diversity of Malagasy Syntomini together with its importance in the protection of the biodiversity of Madagascar are discussed.

Introduction

The biodiversity of Madagascar is characterised by high overall endemism rates of flora and fauna. Despite the fact that an estimated 74% of Malagasy butterfly species live solely there (Vences et al., 2009), endemism of higher taxonomic units within the Lepidoptera fauna is very rare (species- and generic-level endemism predominates), and the few higher-rank exceptions encompass small number of species (Whalleyanidae—2 spp., Callidulidae: Griveaudiinae—3 spp., Hesperiidae: Malazinae—3 spp.; Drepanidae: Nidarini—5 spp., Erebidae: Phryganopterygina—20 spp.) (Lees & Minet, 2003; Zhang et al., 2020; see also Twort et al., 2020). However, the most outstanding example of Lepidoptera endemism on the island are members of the tribe Syntomini. Presently they comprise 99 valid described species in 15 genera, entirely endemic to Madagascar (Viette, 1990; with exclusion of Euchromia spp. in Arctiini, Zenker et al., 2017). Fourteen of these genera (98 spp.) belong to a single evolutionary lineage deriving from a megadiverse radiation, which also gave raise to what have been interpreted as out-of-Madagascar dispersal events, reaching Africa (Pseudonaclia puella (Boisduval, 1847)), Mauritius and even the Palaearctic (genus Dysauxes) (Przybyłowicz et al., 2019; Przybyłowicz et al., 2021, unpublished data). However, since the “Amatidae” monograph of Griveaud (1964), little attention has been paid to the systematics and biogeography of the group, with only a few additional species described by P. Griveaud and P. Viette that were listed in Viette (1990), until the phylogenetic paper of Przybyłowicz et al. (2019). The latter paper and our further examinations show that the present systematic arrangement comprises several genera which are artificial assemblages of not closely related species rather than evolutionary monophyletic units. Thus, the diversity of Malagasy Syntomini demands wider investigation and revision to reveal real relations within the clade, and this we are undertaking.

Here we focus on the Syntomini of the Réserve Spéciale d’Ambohitantely, showing that it appears to be a centre of local richness and endemism for this group, despite its small size (currently around 1,300 hectares, in one large and many small fragments). The Reserve is located in the Central Highlands (usually known as the Central Plateau) of Madagascar. This Plateau covers about 40% of the area of the island, but at the same time is one of the most neglected regions in terms of conservation (Goodman & Raherilalao, 2003; Kull, 2012).

A checklist of Syntomini species of the Reserve is provided for the first time, with the further remarks on each of them. Females of Thyrosticta vestigii Griveaud, 1964 and Maculonaclia tampoketsya Griveaud, 1969 are described, as well as a yellow morphotype of Thyrosticta dilata Griveaud, 1964 which was omitted in its original description. We also redescribe the genitalia of collected species and illustrate them with photographs. These species have already been described and illustrated by Griveaud (1964, 1969), but in a very general and schematic way. This paper is the first of a series planned to adequately describe the syntomine fauna of the island. In terms of diversity, endemism and importance for biodiversity conservation and evaluation this rich radiation may be of similar importance to the lemurs (Mittermeier et al., 2010) but has been neglected in recent surveys.

Materials & Methods

Study area

Research was conducted in the Réserve Spéciale d’Ambohitantely, 80 km north-northwest as measured linearly (though about 130 km by the road) from the capital city Antananarivo (Fig. 1A). The Reserve, situated mostly between 1,250–1,500 m elevation, with the highest point at about 1,670 m, is located in the Central Highlands of Madagascar, in the eastern part of the geological/vegetational formation referred as Tampoketsa (high plateau) d’Ankazobe (Abraham et al., 1996; Goodman, Raherilalao & Wohlhauser, 2018). Its total surface equals to 4,950 ha (5,600 ha in the original creation decree) (Goodman, Raherilalao & Wohlhauser, 2018), and comprises the only significant area of forest in the Ankazobe region (Klein, 2004) and actually one of the last remnants in the Central Highlands at all (Ratsirarson et al., 2003), along with Ankazomivady (Goodman et al., 1998), today very degraded.

Figure 1 Réserve Spéciale d’Ambohitantely.

(A) General localisation on Madagascar, black dot indicates the capital city Antananarivo, green dot—Ambohitantely Reserve. (B) Detailed map of the Reserve, remnant patches of forest marked in green, area of the Reserve hatched, red dot—the light trap setup point ca. 100 m from the forest edge, blue dot—the furthest point on the daytime transect, near the waterfall. (C) General view of the southern part of the Reserve, with the forest covering hilltops (in the middle and on the right side), while in surrounding areas it is constrained to valley bottoms (background), foreground covered with ferns being an important element of herbaceous vegetation in the area. (D) Intermediate zone of shrubby vegetation between grassland (foreground) and forest (background) occurring in some fragments of the Reserve. Map Data: © 2021 Maxar Technologies, © 2021 CNES/Airbus. Photo credit: Marcin Wiorek.

The most up-to-date map of vegetation of the area, including different types of land coverage, based on the analysis of satellite images from 2017 will be published soon by S.M. Goodman (2021, unpublished data). The map prepared by us (Fig. 1B), based on aerial photographs from Google Earth Pro taken in 2016, shows fragmentation of woody vegetation into numerous patches, including areas covered with dense forest, as well as small groups of trees growing in ravines, which are not necessarily comparable in quality and composition to closed-canopy forest. It corresponds with the commonly cited (e.g., Langrand & Wilmé, 1997; Vallan, 2000, 2003) map of Langrand (1995) based on the aerial photographs from 1991. Fragments of forest in the Reserve and in the 10 km wide peripheral zone cover together a calculated 1,302.4 ha and are separated by grasslands and marshy patches (Goodman, Raherilalao & Wohlhauser, 2018; S.M. Goodman, 2021, unpublished data).

Forest in the Reserve is not constrained to valley bottoms as in surrounding areas but covers also hilltops (Fig. 1C). It can be generally classified as medium elevation moist evergreen forest, a formation characteristic of the floristic Central Domain of Madagascar (Gautier et al., 2018), with few variants of vegetation composition identified there, depending on topography and probably corresponding with different stages of its restoration (Goodman, Raherilalao & Wohlhauser, 2018). This type of forest is subject in Ambohitantely to a cool, humid tropical climate, with a high rainfall (around 1,460 mm per year), falling mostly in the warm rainy period lasting for about half a year from October–November to March–April, and high air humidity causing frequent morning mists (Langrand, 2003; Goodman, Raherilalao & Wohlhauser, 2018). High Plateau forests are considered relict, nonetheless grassland is also a naturally occurring vegetation type of the Central Plateau of Madagascar (Solofondranohatra et al., 2020), but nowadays is highly degraded. In Ambohitantely secondary grasslands cover 40% of the protected area (Ratsirarson et al., 2003; Goodman, Raherilalao & Wohlhauser, 2018). The ecotone between forest and grassland is most often sharp (Vallan, 2000), but relatively narrow intermediate zones of secondary shrubby vegetation are also present in some parts of the Reserve (Fig. 1D). Human impact on the environment of the Central Plateau is undeniable (Langrand, 2003; Goodman, Raherilalao & Wohlhauser, 2018), however its extent has been a subject of a great debate, reaching back to the 19th century (Klein, 2004; see also discussion).

Sampling methods and morphological studies

Field collecting was undertaken under the permits Nos. 251/06/MINENV.EF/SG/DGEF/DPB/SCBLF/RECH and 292/19/MEDD/SG/DGEF/DGRNE from Direction Generale de l’Environment et des Forets and Direction de la Gestion des Ressources Naturelles Renouvelables et des Ecosystemes. Material was collected during three visits: 6–7 December 2006, 11–12 December 2019 and 13–16 March 2020, in the southern part of the Reserve (S 18.1969°, E 47.2847°), close to the camp located near the largest patch of forest (S 18.1981°, E 47.2816°). Moths were sampled at night with the use of automatic light traps with UV-A (blacklight) or a 6 W white fluorescent light source, or at a three spectral peak LEPI-LED (Brehm, 2017) inside a reflective screen column, and during the day, between 10 am and 5 pm, by walking slowly through the forest paths and looking for individuals sitting on the upper side of leaves and catching them with a standard entomological net, which is the most efficient method of collecting Malagasy Syntomini (Ł. Przybyłowicz et al., 2021, unpublished data). Day collecting was conducted within a distance of few hours of slow walk from the camp, the final point marked at a cascade near the centre of the largest patch of the forest (Fig. 1B). The light traps were set at dusk (about 6 pm) and left overnight in proximity of the camp site in different habitats: inside the dense forest, as well as among shrubby vegetation on its edges and on sparse trees in grassland surrounding the forest, ca. 100 m from the forest edge (Fig. 1B). In the latter case the light trap was clearly visible to moths flying out from the forest. Collected moths were killed with ethyl acetate and pinned on standard entomological pins. Individuals were spread after legs were sampled for molecular studies and photographed with a Canon 70D camera before the further examination. The images were adjusted with Adobe Photoshop. Specimens are deposited in the collection of ISEA PAS, Kraków, Poland, accession numbers of the specimens are provided in Table S1.

Genitalia were dissected from one specimen of each sex of collected species, except for males of Tsarafidynia perpusilla, where two slides of the same sex were prepared. Abdomens were macerated in 10% KOH in a water bath, then the genitalia were stained with chlorazol black, embedded in Euparal (Agar Scientific, Essex, UK) and mounted on slides. Photographs of the genitalia were taken with the use of a stereoscope microscope Leica S9i system. Images were adjusted with the Adobe Photoshop programme.

The general morphological terminology follows Miller (1991), and for genitalia we refer to Koda (1987). Measurement of forewing length (in mm) was taken with the use of a digital caliper from base to apex of the wing.

We summarise our results with updated data published on distribution and ecology of Syntomini species occurring in the Réserve Spéciale d’Ambohitantely. Official names of protected areas mentioned in the text follow Goodman, Raherilalao & Wohlhauser (2018). The most important collections of Malagasy Syntomini, including type specimens of majority of the species, are deposited in three institutions: MNHN in Paris, NHMUK in London and PBZT in Antananarivo. However, we may have missed some information, as in the latter collection, red paratype labels are pinned under the main labels, thus in available photographs were visible partially, if at all. Moreover, some species have more specimens labelled as paratype than designated in Griveaud (1964), even taking into consideration only London and Paris collections, where the photographed labels are clearly visible. These last collections were inaccessible for examination at the time of writing (due to COVID-19). Thus, to make some morphological and taxonomical remarks and to confirm localisation of type specimens of species that are dealt with in the paper, we relied on photographs of specimens and genital slides taken by ŁP in MNHN in 2015 and in PBZT in 2019. Detailed photographs of type specimens and their labels from NHMUK are available on AfroMoths (De Prins & De Prins, 2011) and from MNHN are available in the Museum online database of the Lepidoptera collection (https://science.mnhn.fr/institution/mnhn/collection/el/item/search).

Molecular studies

For molecular investigation, two legs from each specimen were sampled. Isolation of genomic DNA was done with the NucleoSpin Tissue kit (Machery-Nagel, Germany), following the manufacturer’s protocol. Sequences of the first part of the mitochondrial gene cytochrome c oxidase subunit I (COI) were obtained with the use of HCO/LCO primers pair hybridised with the universal primer pair T7/T3, described by Wahlberg & Wheat (2008). PCR was done with the use of hot-start ready PCR mix (StartWarm HS-PCR Mix; A&A Biotechnology, Poland), protocol followed manufacturer’s instructions. Obtained sequences were compared with chromatograms, aligned manually with a template sequence in BioEdit software (Hall, 1999). Ambiguous sites were coded in accordance to the IUPAC nucleotide code.

Prepared sequence files were managed with VoSeq database (Peña & Malm, 2012). Sequences were analysed in a Maximum Likelihood framework in IQ-TREE (Nguyen et al., 2015) on the web server (Trifinopoulos et al., 2016) with 1,000 replications of Ultrafast Bootstrap (Minh, Nguyen & von Haeseler, 2013). The p-distance between barcode sequences was calculated in MEGA X (Kumar et al., 2018). The outgroup sequence of Fletcherinia decaryi Griveaud, 1964 (GenBank accession code MK158546) was obtained from the study of Przybyłowicz et al. (2019). DNA sequences are deposited in GenBank (MW817635–MW817665) and accession codes are provided in Table S2.

Results

Checklist of Syntomini of Réserve Spéciale d’Ambohitantely and general remarks

During three expeditions to Ambohitantely 58 specimens of Syntomini belonging to seven species in four genera were collected. In total, the fauna of Syntomini of the Reserve comprises eight species, of which two are recorded for the first time (marked with ‘!’). At the current stage of knowledge on distribution of Malagasy Syntomini, five species appear to occur only in this area (marked with ‘*’):

*Maculonaclia altitudina Griveaud, 1964

!Maculonaclia ankasoka Griveaud, 1964

*Maculonaclia brevipenis Griveaud, 1964

*Maculonaclia tampoketsya Griveaud, 1969

*Thyrosticta dilata Griveaud, 1964

*Thyrosticta vestigii Griveaud, 1964

Tritonaclia stephania (Oberthür, 1923)

!Tsarafidynia perpusilla (Mabille, 1880)

Three species: Maculonaclia ankasoka, Tritonaclia stephania and Tsarafidynia perpusilla are known from several localities in central, eastern and southern parts of Madagascar (Fig. 2). Most of them are located near or within protected areas, as indicated in the text. In all these areas, as well as in Ambohitantely, the dominant type of vegetation is medium elevation moist evergreen forest (sensu Gautier et al., 2018). The above three more widely distributed species occur between 800 and 1,600 m elevation, with Ambohitantely being the highest observed locality in all the cases.

Figure 2 Records of widely distributed species occurring in the Réserve Spéciale d’Ambohitantely.

(A) Maculonaclia ankasoka. (B) Tritonaclia stephania. (C) Tsarafidynia perpusilla, red dot indicates Ambohitantely, blue dots—remaining localities listed in the text.

Type specimens of the most of Malagasy Syntomini species, including all described by P. Griveaud, are deposited in three collections, but in Griveaud (1964) only details concerning holotypes and allotypes were given. Generally, holotypes as well as part of paratypes are housed in MNHN Paris, remaining paratypes are in NHMUK London and in PBZT Antananarivo.

Specimen data, descriptions and remarks on the species

Maculonaclia altitudina Griveaud, 1964

Distribution

Until now recorded only in the Réserve Spéciale d’Ambohitantely.

Remarks

Maculonaclia altitudina is the only Syntomini species occurring in the Reserve, which was not collected during our study. This species is known only from type series, consisting of the male holotype and seven paratypes, all collected in the Reserve by A. Robinson in May 1961 at an elevation of 1,550 m. The female remains unknown (Griveaud, 1964). The holotype and three paratypes are deposited in MNHN, and one paratype is in NHMUK. A further three specimens are in PBZT, and their collecting data labels are identical to those of specimens in Paris and London, thus they probably are the remaining paratypes. Among specimens labelled as paratype of Maculonaclia altitudina in Paris there is one additional specimen, undoubtedly belonging to the species Maculonaclia brevipenis, which is similar in general appearance, but distinctly differs in details of forewing pattern. As discussed below, the type series of Maculonaclia brevipenis in Paris contains many more specimens labelled as paratypes than stated by Griveaud (1964), and all the specimens of both species were collected in the same place, at the same time and by the same collector.

All the known specimens of Maculonaclia altitudina were collected in May, in the cool dry period.

Maculonaclia ankasoka Griveaud, 1964 (Figs. 2A, 3A, 4A)

Figure 3 Syntomini of Ambohitantely, resting posture.

(A) Maculonaclia ankasoka, female. (B) Maculonaclia brevipenis, male. (C) Maculonaclia tampoketsya, female. (D) Thyrosticta vestigii, female. (E) Thyrosticta dilata, black morphotype, male. (F) Thyrosticta dilata, yellow morphotype, male. (G) Tritonaclia stephania, male. (H) Tsarafidynia perpusilla, male.

Figure 4 Female genitalia of syntomini of Ambohitantely.

(A) Maculonaclia ankasoka. (B) Maculonaclia tampoketsya. (C) Thyrosticta vestigii.

Material (1 specimen). 1♀, 8.xii.2006, Ankazobe District, Ambohitantely Reserve (1,600 m), lgt. Ravo Ranaivosolo.

Distribution (Fig. 2A)

The species is recorded for the first time from the Ambohitantely Reserve. Until now it has been known from four localities given by Griveaud (1964) (from north to south): Périnet [=Andasibe], Ankasoka and Sandrangato—all three located close to each other in the area of the southern parts of the Réserve de Ressources Naturelles du Corridor Ankeniheny-Zahamena and Parc National d’Analamazoatra (on the map all three marked as one point); Tsarafidy—a few kilometres W from Parc National de Ranomafana and about 32 km NNE of Fianarantsoa. It occurs between 900 and 1,600 m elevation. Ambohitantely is so far the northernmost locality.

Redescription of female genitalia (Fig. 4A)

Papillae anales subtriangular with rounded protrusion at base of dorsal margin, covered with short erect setae, much denser on the protrusion; dorsal and ventral pheromone glands present in form of very narrow, elongate, not anastomosing membranous tubes; apophyses posteriores almost as long as papillae anales, straight and narrow, needle-like; apophyses anteriores of similar shape and size as apophyses posteriores; ostium bursae rounded; antrum well developed, sclerotised, cylindrical, slightly longer than wide; ductus bursae membranous, slightly widening towards corpus bursae, terminal portion with sublateral diverticulum directed distally, from which narrow, membranous ductus seminalis originates; corpus bursae forming a membranous, oval pouch bearing indistinct, irregular zones of minute, diffuse scrobinations; central portion with a pair of signa in form of short, parallel ridges consisted of tiny subtriangular sclerotised plates, leaning on each other; along a longitudinal axis designated by the signa, scrobinations are slightly strongly articulated.

Remarks

Male and female genitalia were described and illustrated by Griveaud (1964: Figs. 87–90). Corpus bursae is depicted to possess scrobinations only in the rhomboidal areas surrounding each of two signa. In the genital slide of the allotype (MNHN) these areas are indeed more prominent than in the slide prepared from our specimen (Fig. 4A), where minute scrobinations are diffuse over whole corpus bursae, and only slightly larger around signa. It could be a matter of intraspecific variation, but also an effect of different staining technique, as slides of Griveaud are prepared with eosin, whereas our ones with chlorazol black. This issue needs further examination in the future on larger series of specimens.

Type series of Maculonaclia ankasoka designated by Griveaud (1964) is given to comprise 10 specimens: male holotype, and nine paratypes (one male and eight females of which one labelled as allotype). The holotype and the male paratype were collected by P. Griveaud in November 1956 in Ankasoka at an elevation of 1,000 m (however original label of the holotype says “1,130 m”), the allotype and the remaining female paratypes in February 1961 at Périnet, elevation 900 m (Griveaud, 1964). The holotype, allotype and three other paratypes are deposited in MNHN. In Paris are also other seven specimens: four collected by P. Griveaud and R. Vieu in 1956 and three collected in 1959, 1963 and 1964 by P. Viette (detailed collecting data illegible in the photographs). A further about 30 specimens determined as Maculonaclia ankasoka are in the PBZT collection, collected mostly by P. Griveaud. Three of them are most probably remaining paratypes, as their labels agree with data given by Griveaud (1964).

Maculonaclia brevipenis Griveaud, 1964 (Figs. 3B, 5A)

Figure 5 Male genitalia of Syntomini of Ambohitantely.

(A) Maculonaclia brevipenis. (B) Maculonaclia tampoketsya. (C) Tsarafidynia perpusilla. (D) Thyrosticta dilata, yellow morphotype. (E) Thyrosticta dilata, black morphotype. (F) Tritonaclia stephania.

Material (7 specimens). 2♂♂, 12.xii.2019, Ankazobe District, Ambohitantely Reserve (1,600 m), S 18.1969°, E 47.2847°, lgt. ŁP; 5♂♂ as above but 14–15.iii.2020 (all collected by netting at day).

Distribution

Until now recorded only in the Réserve Spéciale d’Ambohitantely.

Redescription of male genitalia (Fig. 5A)

Tegumen narrow, moderately sclerotised, almost completely fused with vinculum; uncus elongate, dorso-ventrally flattened, slightly concaved in ventral surface; of the same width up to sharply narrowed, ventrally incurved hook-like tip; dorsally covered with erect setae, longer in basal portion; vinculum narrow, produced with a prominent saccus of triangular shape; juxta well developed, divided into transverse ventral plate and a pair of lateral, rectangular plates; valva approximately the length of uncus with terminal half of triangular shape; costa evenly convex, widely folded towards inner zone; tiny, tooth-like protrusion in the 1/3 of folded costal margin; saccular margin shallowly sinusoidal; margins and some regions of internal and external surface with short erect setae; aedeagus weakly sclerotised, short, tubular, slightly narrowing towards apex; vesica membranous, bag-like, with four small sclerotised plates of irregular shape in latero-distal portion.

Remarks

Male genitalia were described and illustrated by Griveaud (1964: Figs. 95–97). Figures show valva with sharply terminated apex, narrow elongate saccus, and vesica was uneverted. In fact the valva is dully terminated and saccus is triangular, but not elongate, vesica as described above.

Until now the species has been known from the type series, according to Griveaud (1964) consisting of the male holotype and two paratypes, collected by A. Robinson in May 1961 at an elevation of 1,550 m. The female remains unknown. However, in MNHN, except the holotype, are deposited 10 specimens marked as paratypes, labelled with identical collecting data as given above. Further two paratypes are deposited in NHMUK. Another 13 specimens with identical labels are in PBZT collection, at least one of which is also labelled as a paratype, because a fragment of a red label is visible from under the collecting data label.

All the known specimens were collected in December, March and May, during the warm rainy period and at the beginning of the cool dry period.

Maculonaclia tampoketsya Griveaud, 1969 (Figs. 3C, 4B, 5B, 6)

Figure 6 Phylogenetic tree based on a Maximum Likelihood analysis of the barcode region.

Yellow dots indicate the yellow morphotype of T. dilata, black dots—black morphotype. Node labels—ultrafast bootstrap values (see Materials & Methods), scale bar—number of substitutions per site.

Material (2 specimens). 1♂, 11–12.xii.2019, Ankazobe District, Ambohitantely Reserve (1,600 m), S 18.1969°, E 47.2847°, lgt. ŁP; 1♀ as above (all attracted by light).

Distribution

Until now recorded only in the Réserve Spéciale d’Ambohitantely.

Redescription of male genitalia (Fig. 5B)

Tegumen completely fused with vinculum, very narrow, moderately sclerotised, with a pair of prominent, sharp, claw-like protrusions directed ventrally, close to the uncus base; uncus large, elongated, bent ventrally, laterally flattened, with tiny spike-like protrusion at the tip; basal half with numerous long setae; saccus short, terminated with tiny, narrow protrusion; valva elongate, reaching almost to uncus tip, narrowed terminally into sclerotised, hook-like process slightly curved ventrally; costal margin widely sclerotised, concaved submedially, with some undulation in its basal portion; concavity marked with a narrow, membranous, joint-like articulation; sacculus sclerotised, reaching till 2/3 of valva length, with short erect setae, extending beyond; a short spike-like protrusion at dorsoterminal margin beyond sacculus; central inner portion of valva membranous; aedeagus tubular, widened subbasally, slightly bent dorsally in distal portion; vesica in form of membranous tube evenly widened in proximal 2/3 of its length, bearing a dense bunch of elongate, needle-like cornuti in terminal portion.

Remark: The short spike-like protrusion at dorsoterminal margin beyond sacculus visible only on right valva. Left valva with indistinct convexity.

Description of female (Fig. 3C)

Head. Proboscis well developed, brown, apex and base pale brown; frons pale yellow, with longitudinal ochraceous stripe from clypeal portion towards second third; vertex ochraceous with admixture of pale yellow scales, lateral margins yellow, ochraceous stripe between scapi; palpi three-segmented, porrect, yellow, ventrally with elongate scales, dorsally with admixture of ochraceous scales, terminal palpomere dorsally entirely ochraceous; antennae filiform, ochraceous with admixture of creamy scales, except terminal, dark ochraceous quarter.

Thorax. Patagia of piliform scales, submedially ochraceous with tiny yellow spot in central portion, laterally pale yellow; tegulae pale yellow with elongate scales almost piliform in distal portion, terminally with admixture of ochraceous; subventral zone ochraceous; mesothorax ochraceous, medially with longitudinal narrow yellow stripe and yellow spot in disto-median portion; metathorax ochraceous; ventral portion of pleurites ochraceous, with yellow blotches at base of coxa; foreleg: pale yellow, epiphysis absent; midleg: pale yellow, tibia with one pair of terminal spurs of similar length; hindleg: coxa and femur pale yellow; remaining parts of the hindleg unavailable.

Abdomen. Ochraceous, distal margin of each segment with yellow stripe.

Forewing. Length of costa 11 mm (n = 1); upperside background ochraceous, with short, yellow, narrow streak along proximal portion of dorsum and additional 5 pale yellow to creamy blotches of subrectangular shape and similar size: 1 at basal 2 at medial and 2 at distal portion of wing; basal one elongate, from costal margin to the half of the wing width, with a prominent narrow projection towards wing base on R vein; first medial one of rectangular shape, form costal margin to hind margin of DC; second medial one of irregular shape, from cubital vein, widening towards termination before inner margin; first distal one elongate, from costal margin to M3, constricted in medial portion along M1; second distal one below CuA1, of irregular shape, separated from outer margin by narrow ochraceous stripe; underside with the same pattern, with addition of zone of scattered pale yellow scales between the basal blotch and 1A + 2A; cilia ochraceous.

Hindwing. Elongate, reaching about half of forewing; basal portion yellow, reaching to the baso-distal angle and to 3/4 of the length of costal margin, with large, 8-shaped elongate ochraceous blotch, originating from the wing base and including most of DC, but not reaching to its outer margin nor the costal margin of the wing; outer area ochraceous; underside pattern the same but lateral portion of brown blotch reaches the costal margin; piliform scales along wing margins, longer on baso-distal margin; frenulum present.

Female genitalia (Fig. 4B)

Papillae anales semicircular, covered with short, dense, erect setae; dorsal pheromone glands present in form of narrow, rather stright, not anastomosing tubes, of about three lengths of apophyses posteriores; apophyses posteriores strongly sclerotised, straight and narrow, needle-like; apophyses anteriores in form of subtriangular, short lobes, half of the length of apophyses posteriores; 7th and 8th segments heavily sclerotised; 7th sternite wide and narrow with shallowly concave distalomedian margin and a pair of shallow depressions at anterolateral corners; 8th sternite with distinct, expanded, subtriangular wrinkled cavities at anterolateral margin; posterior margin in form of prominent, sclerotised ridge provided medially with deep, U-shaped slit connected with ostium bursae by well-defined concavity of parallel margins; ostium bursae rounded, strongly sclerotised; antrum well developed, strongly sclerotised, distinctly bent distally to the left (according to body axis); ductus bursae strongly bent to the right towards the medial axis, membranous, of length of antrum, slightly widened terminally, with plicae in form of longitudinal parallel ridges; ductus seminalis from anterior portion of antrum just below the ostium; corpus bursae membranous, pear-shaped, bearing extensive, irregular zone of minute, diffuse scrobinations; central portion of which with a pair of spiny signa, proximal one elongate, terminal one rounded with longer spines than in proximal one.

Remarks

We collected one specimen of each sex, thus in this case we could confirm that they are conspecific not only by morphological, but also molecular examination (Fig. 6). The genetic distance between barcode sequences (p-distance) is 0% (Table S2).

The male was illustrated as a line drawing by Griveaud (1969: pl. I, Fig. B), but in the figure caption was referred as “Melanonaclia tampoketsya”, which is certainly an unintended error, as in the description this species is explicitly attributed to the genus Maculonaclia and to the section of Maculonaclia ankasoka established within the genus by Griveaud (1964). The male genitalia were described and illustrated in the same paper (Griveaud, 1969: Figs. 9–12), but with an uneverted vesica of the aedeagus, which is described above.

Until now the species has been known only from male holotype collected by P. Griveaud in April 1967, deposited in MNHN (Griveaud, 1969). In the male the general body colouration and pattern are similar to the female, with differences listed below: eyes much larger, with tuft of yellow scales at the eye margin, below scapus; frons narrower; vertex uniformly ochraceous, with yellow lateral margins and small yellow spot in central part, over axis between scapi; antennae serrate, shaft dorsally golden-yellowish, each pectine with golden-yellowish lobe directed downwards, cowered with short, dense, erect setae; hindleg tibia possess one pair of spurs of slightly uneven length; all legs have well-developed arolium; retinaculum is present. Male palpi are similar to female, i.e., yellow, dorsally with admixture of ochraceous scales, terminal palpomere dorsally entirely ochraceous, but in original description (Griveaud, 1969) are referred as entirely ochraceous, which can be intraspecific variation and needs to be revised in MNHN collection.

All the known specimens were collected in April and December, thus both in cool dry and warm rainy period.

Thyrosticta dilata Griveaud, 1964 (Figs. 3E, 3F, 5D, 5E and 6)

Black morphotype

Material (23 specimens). 12♂♂, 11–12.xii.2019, Ankazobe District, Ambohitantely Reserve (1,600 m), S 18.1969°, E 47.2847°, lgt. ŁP; 2♂♂ as above but 11.xii.2019; 6♂♂ as above but 13–15.iii.2020 (all above attracted at light); 2♂♂ as above but 14–15.iii.2020; 1♂ as above but 12.xii.2019 (latter 3 collected by netting at day).

Yellow morphotype

Material (22 specimens). 15♂♂, 11–12.xii.2019, Ankazobe District, Ambohitantely Reserve (1,600 m), S 18.1969°, E 47.2847°, lgt. ŁP; 5♂♂ as above but 13–15.iii.2020 (all above attracted at light); 1♂ as above but 12.xii.2019; 1♂ as above but 14–15.iii.2020 (latter 2 collected by netting at day).

Distribution

Until now recorded only in the Réserve Spéciale d’Ambohitantely.

Taxonomic status

This species is represented by two morphotypes, described in detail below. Despite variation within and clear differences between them, our morphological and molecular results confirm that they belong to the same species. There is no difference in male genitalia (Figs. 5D and 5E), also p-distance between barcode sequences of specimens varies from 0–0.2% with no regard to morphotypes (Table S2), and all the specimens represent a single clade on the tree (Fig. 6).

Description of the yellow form (Fig. 3F)

Head. Proboscis well developed, black, with ochraceous-yellowish apex; frons yellow, few pale ochraceous scales close to eye margin; vertex yellow, with black stripe between scapi and ochraceous longitudinal spot in median part; palpi porrect, terminally curved downward; palpomeres of comparable length, elongated, at least three times longer than wider; first two palpomeres yellow, with ochraceous scales on dorsal part; first with piliform scales on ventral part; third palpomere ochraceous, with admixture of yellow scales; antennae bipectinate; Scapus yellow ventrally, ochraceous dorsally; shaft black ventrally, dorsally yellow at base, distally from base with admixture of ochraceous scales, increasing towards entirely ochraceous apex; pectines black with numerous dense, short, erect setae; on each pecten 3 yellowish-ochraceous setae of uneven length, apical one the longest and most visible.

Thorax. Patagia black with admixture of ochraceous and yellow scales in lateral parts; tegulae yellow, black basally; mesothorax yellow with dark ochraceous central portion; metathorax with elongated scales, medially yellow, laterally ochraceous; ventral pleurites yellowish-ochraceous with yellow spots at base of mid and hind coxa; foreleg: coxa ochraceous with pale yellow stripe on lateral and distal margins; femur and tibia ochraceous medially, yellow laterally; epiphysis ochraceous, reaching 3/4 the length of tibia; tarsus ochraceous, segments 1–3 partially pale yellow; midleg: coxa pale ochraceous; femur yellow, ochraceous terminally; tibia pale ochraceous medially, pale yellow laterally, one pair of pale yellow terminal spurs of uneven length; tarsus pale yellow with pale ochraceous admixture; hindleg: coxa pale ochraceous; femur pale yellow with pale ochraceous terminal portion; tibia pale yellow with pale ochraceous admixture, two pairs of pale yellow spurs; tarsus pale ochraceous with pale yellow admixture.

Abdomen. Ochraceous, each segment with yellow, differently expressed distal margin, gradually broadened towards the abdomen termination, hardly visible on the first tergite.

Forewing. Upperside background ochraceous, with narrow, yellow streak from wing base to its half between costal margin and Sc, and additional four yellow blotches of different shape: two at medial and two at distal portion of wing; the largest first medial blotch in DC, U-shaped, fusing with the yellow streak; second rounded, between CuA2 and 1A + 2A; third one in apical region, round, with comma-shape projection towards first medial one; fourth one 8-shape, between M2 and CuA1; underside with the same pattern; inner margin with piliform scales; cilia ochraceous; retinaculum present.

Hindwing. Elongate, reaching beyond half of forewing; basal portion including DC, half of the costal margin and basal portion of hind margin-yellow, and in central part forming a round projection into outer ochraceous zone; underside pattern the same, with ochraceous costal margin, broadened at wing base, and tiny protrusion from the margin towards central part of yellow zone; hind and outer margins with elongated scales, piliform at the wing base; frenulum present.

Redescription of male genitalia (Figs. 5D and 5E)

Tegumen moderately sclerotised, widened in dorsal portion, laterally narrowed, almost completely fused with vinculum; uncus narrow, elongate, dorsally with long, erect setae; slightly constricted in distal third, terminated in form of a sclerotised, sharp, dorso-ventrally flattened tip; vinculum very narrow, U-shaped, without produced sacculus; valva moderately elongated, subtriangular, narrowed till dull apex; terminal half including margins with short, erect setae, inner portion with shallow, longitudinal convexity; costal margin of sinusoidal shape, terminally with spike-like inwards curved protrusion, outer margin shallowly concave in distal portion; saccus not developed; aedeagus moderately elongate, of approximately equal width, L-shaped; vesica membranous, elongate, tubular; short subbasal portion distinctly bent parallel to aedeagus base; its left lateral zone with pocket-like diverticulum provided with a indistinct field of minute scrobinations, opposite membranous wall without diverticulum but with more extensive field of distinctly thicker scrobinations; remaining portion of vesica delicately spiral, provided with a belt-like longitudinal zone of granular sclerotisations covering less than a half of the vesical membrane circumference.

Remarks

The specimens representing the yellow morphotype strongly resemble Thyrosticta vieui Griveaud, 1964, especially in the black-yellow striped abdomen and the general pattern of the forewing. The main differences are: (i) shape of the basal blotch of the forewing, in Thyrosticta dilata forming a narrow, yellow streak between costal margin and Sc, while in Thyrosticta vieui present as a wider, irregular, suboval blotch with fuzzy margins, close to the costal margin; (ii) shape of the projection of yellow blotch of the hindwing, distinctly narrower in Thyrosticta vieui than in Thyrosticta dilata.

The black morphotype distinctly differs from the yellow one in the characters listed below (Figs. 3E and 3F): head (including palpi and antennae) and thorax (including patagia) are entirely black, tegulae yellow with piliform scales. Legs are fully dark ochraceous, including spurs. Forewing has the same shape and pattern as in yellow form, but elongate streak along costal margin is always absent. Abdomen is entirely black dorsally and ventrally, first tergite possesses elongated black scales.

As already mentioned, no intermediate form of Thyrosticta dilata has been detected, however both morphotypes exhibit internal variation in the colouration described below.

In the yellow morphotype (n = 22) ochraceous scales on frons, close to eyes margin, are absent in some specimens. The ochraceous stripe on vertex varies from a very narrow band to a globular blotch, reaching or not to the black stripe between antennae. The elongate streak on costal margin of forewing reaches half of the wing and fuses with the second U-shaped blotch or terminates before. In some specimens also a comma-shape projection of apical blotch reaches close to or fuses with the DC blotch, up to the fusion of these three blotches, creating a yellow stripe along costal margin fused with them. When the wing pattern is strongly developed, the U-shaped (the largest) and round (the second) blotches nearly touch each other, but never fuse.

In the black morphotype (n = 23) some specimens have a general body colouration of dark ochraceous rather than blackish. In some specimens with strongly developed wing pattern the U-shaped (the largest) and round (second) blotches nearly touch each other, up to their fusion.

The species has until now been known only from the type series designated by Griveaud (1964), stated to consist of the male holotype and four paratypes. The female remains unknown. The holotype was collected by P. Griveaud on 27.xii.1956 at an elevation of 1,600 m. Paratypes are said to have been collected in May 1961 and to have the same provenance and collector as the holotype (Griveaud, 1964), but according to their labels, all the five specimens from May 1961 were collected by A. Robinson at an elevation of 1,550 m, not by P. Griveaud at 1,600 m.

The holotype and one paratype are deposited in MNHN, another paratype is in NHMUK. A further three specimens with labels identical as those of the paratypes in Paris and London are in PBZT, thus most probably among them are the remaining two paratypes. In PBZT there are also an additional four specimens, collected in April 1967 by P. Griveaud (two specimens), in October 1974 by A. Peyrieras (one specimen) and in the 1970s (one specimen, exact year and name of collector illegible in the photograph). All of them were collected in the area of Tampoketsa d’Ankazobe as well, however the specimen from 1974 remains uncertain because of an illegible locality on the label, except “central Madagascar”.

For the reason given below, we assume that Griveaud was aware of the intraspecific variation when describing the species, but for some reason omitted it. The original description and colour illustration (Griveaud, 1964: pl. I, Fig. 60) refer to the black morphotype. However, the holotype deposited in MNHN represents the yellow morphotype, while the paratype in the same collection belongs to the black one. The genitalia were described and illustrated in Griveaud (1964: Figs. 224–226), but with an uneverted vesica on the aedeagus, which is described above.

As indicated in the Materials section, almost all of the fresh specimens of Thyrosticta dilata were collected at light traps with both UV or non-UV white light sources, which allowed us to obtain a series of well-preserved specimens. According to our observations, this is rather exceptional among Malagasy Syntomini, although DCL has observed it for some members of genera Thyrosticta and Tritonaclia at other sites. As a general rule, syntomines are attracted to light rather rarely and usually just in small numbers, which makes day netting the most efficient collecting method for the vast majority of taxa.

Thyrosticta vestigii Griveaud, 1964 (Figs. 3D and 4C)

Material (1 specimen). 1♀, 12.xii.2019, Ankazobe District, Ambohitantely Reserve (1,600 m a.s.l.), S 18.1969°, E 47.2847°, lgt. ŁP (collected by netting at day).

Distribution

Until now recorded only in the Réserve Spéciale d’Ambohitantely.

Description of female (Fig. 3D)

Head. Entirely blackish ochraceous, including palpi and antennae; palpi projected downward; proboscis well developed; antennae filiform, flagellum with numerous short, erect setae.

Thorax. Concolorous with head both dorsally and ventrally, including patagia and filiform tegulae; metascutellum with partially filiform scales; legs entirely blackish ochraceous, with exception of paler epiphysis on foreleg; mid and foreleg tibia with one pair of terminal spurs.

Abdomen. Entirely blackish ochraceous dorsally and ventrally, with admixture of piliform scales.

Forewing. Length of costa 7 mm (n = 1); upperside blackish ochraceous, with two partially fused yellow blotches; first one prominent, reaching from the wing base up to half of the wing length terminating at DC outer margin; costal portion along R stem with indistinct, shallow, concavity in its half-length; opposite margin in proximal part along narrow ochraceous streak of inner margin of wing, in distal part directed to a right-angle-shaped terminating blotch; second blotch in postdiscal zone, of dumbbell-shape, fusing narrowly in inner posterior angle with the tip of first blotch; cilia and scales along inner margin elongate concolorous with background; underside pattern the same.

Hindwing. Oval, elongated, reaching beyond the half of forewing; basal part including DC with yellow oval zone, reaching to the baso-distal angle and beyond the half of the costal margin; outer zone brown, with narrow brown margin along costa; underside pattern the same, with addition of short brown protrusion from brown costal margin towards central part of yellow zone; elongated scales on outer and hind margins, with dominance of piliform scales close to wing base; frenulum present.

Female genitalia (Fig. 4C)

Papillae anales subtriangular, covered with erect setae, much denser and longer in ventral portion; dorsal pheromone glands well developed, in form of four very narrow, elongate, twisted, rarely anastomosing membranous tubes; two sublateral, much longer than two submedial; apophyses straight and narrow, needle-like; posteriores as long as papillae anales, anteriores slightly shorter; ostium bursae membranous, with lateral projections of subtriangular shape, covered with minute scrobinations; antrum well developed, wide and at least two times longer than wide, plain, weakly sclerotised; proximal margin of 7th segment laterally with symmetrical pocket-like cavities covered with scales; ductus bursae membranous, constricted in middle portion, inner wall in form of sclerotised plate, outer one membranous with well defined, longitudinal, parallel plicae; corpus bursae in form of membranous, elongate pouch, entirely covered with conspicuous plicae in form of longitudinal, parallel ridges; signum singular, prominent, forming a strongly sclerotised, narrow, elongate longitudinal buckle, located at laterobasal portion of corpus bursae; initial portion of signum widened and folded inwards, palm-shaped, formed of four subtriangular plates of different size; terminal portion straight, reaching half of corpus bursae, with longitudinal row of spine-like protrusions of different length directed inwards corpus bursae, and row of few scrobinations on outer surface; below the signum, in terminal third of corpus bursae three tiny spine-like scrobinations directed inwards; ductus seminalis narrow, from membranous diverticulum in basal portion of corpus bursae.

Remarks

Body colouration and wing pattern of female is generally the same as in male. All the blackish-ochraceous body parts have golden-yellowish reflections.

The male and its genitalia were described and illustrated in Griveaud (1964: Pl. I, Fig. 56; Figs. 208–210).

Until now the species has been known only from male holotype and two paratypes collected by A. Robinson in May 1961 at an elevation of 1,550 m (Griveaud, 1964). The holotype and one paratype are deposited in MNHN. One specimen has an identical label, thus being most probably the second paratype is in PBZT. In the latter collection there is also one additional worn specimen, labelled as collected in “Tampoketsa d’Ankazobe” in October 1974 by A. Peyrieras. Thus, this specimen was collected somewhere around the Ambohitantely Reserve, and the species remains endemic to the area.

All the specimens known to us were collected in October, December and May, so both during the warm rainy period and at the beginning of the cool dry period.

Tritonaclia stephania (Oberthür, 1923) (Figs. 2B, 3G, 5F)

Material (1 specimen). 1♂, 11–12.xii.2019, Ankazobe District, Ambohitantely Reserve (1,600 m a.s.l.), S 18.1969°, E 47.2847°, lgt. ŁP (attracted by light).

Distribution (Fig. 2B)

This species is known from the few localities given by Griveaud (1964) (from north to south): “Réserve Naturelle III”—present Parc National de Zahamena; Réserve Spéciale d’Ambohitantely; La Mandraka—ca. 10 km S of Paysages Harmonieux Protégé du Complexe Anjozorobe-Angavo; “Ampolomita”—east of Belanitra (for details see Griveaud, 1957); Tsarafidy/Ankafina—about 32 km NNE of Fianarantsoa; “préfecture de Fianarantsoa” (not shown on the map, see remarks). Occurs between 800 and 1,600 m elevation, Ambohitantely being the highest recorded locality.

Redescription of male genitalia (Fig. 5F)

Tegumen narrow, moderately sclerotised, not fused with vinculum, with a pair of lateral, flattened protrusions, slightly curved dorso-distally and densely covered with short setae; uncus base trapezoidal, recessed into disto-dorsal wall of tegumen, surrounded laterally by short tegumen arms; uncus prominent, bent ventrally, of arrowhead shape; narrowed in medial portion, with dorsal, longitudinal rib in distal portion; subdorsally covered with long, erect setae directed laterally; apex bulbous with round, concave tip; vinculum narrow, tendril-like laterally, produced medially into short, triangular, sharply terminated saccus; valva elongate, of claw-like shape, narrowed in terminal half with sharp tip slightly curved inward, terminally with few erect setae; costa at 1/3 of its length with nodular protrusion, folded towards central, membranous part of valva; sacculus sclerotised, reaching to the half of valvae length, with erect setae on margin then in form of membranous, sclerotised, textured lobe, reaching nearly till the end of valva, on outer margin with thin, erect setae; aedeagus massive, tubular, widened basally, gradually narrowing towards termination; vesica membranous, tubular, widened in basal portion, with longitudinal row of eight sharp, thick, spike-like cornuti bent towards the base of vesica; terminal portion with a pair of adhered to each other, sclerotised plates of subtriangular shape of which the outer one much larger than the inner one.

Remarks

Tritonaclia stephania was originally described and illustrated by Oberthür (1923: 135, pl. 566, Fig. 4882) from southern Madagascar (Sud de Madagascar Reçu de M. Lamberton en Avril 1922), but he did not mention number nor sex of specimens. Generally, Charles Lamberton block locality labels, and especially this one, are unreliable, even as to the part of the island (Viette, 1962: 15) as also seen for some butterflies so labelled which are expected only to occur in the North (D.C. Lees, 2020, personal observations). However, the male specimen deposited now in NHMUK has a label indicating a collecting locality agreeing with that given in the original description and another label with information that the specimen was a model for the illustration in the original description. Griveaud (1964: 80, Figs. 184–187) described the male and female genitalia and designated the aforementioned male specimen (NHMUK010620988) from London as lectotype, and another specimen (female) from MNHN, which he apparently recognized as a part of the type series, as “neallotype”. According to the labels both specimens were obtained from Ch. Lamberton in 1922, but collecting dates are unknown. However, Griveaud (1964) probably recognized the locality “Fianarantsoa” where the “neallotype” was collected as a very general area and gave “préfecture de Fianarantsoa”, which was a larger unit of the former administrative division of Madagascar. For this reason, the locality is not shown on the map (Fig. 2B), but the southernmost locality Tsarafidy is ca. 30 km NE of the city of Fianarantsoa, thus this exclusion does not change the general range of the species significantly.

There are further 21 specimens deposited in MNHN, collected by P. Griveaud, P. Soga and R. Vieu, and 20 specimens in PBZT, collected mostly by P. Griveaud.

Tsarafidynia perpusilla (Mabille, 1880) (Figs. 2C, 3H, 5C)

Material (1 specimen). 1♂, 14-15.iii.2020, Ankazobe District, Ambohitantely Reserve (1,600 m a.s.l.), S 18.1969°, E 47.2847°, lgt. ŁP (collected by netting at day).

Distribution (Fig. 2C)

The species is recorded for the first time from the Ambohitantely Reserve; it has so far been recorded from three localities (Griveaud, 1964) (from north to south): Antananarivo; Tsarafidy forest (erroneously written as “Tsarafify”)—a few kilometres W from Parc National de Ranomafana; “sous-préfecture de Midongy du Sud”—currently district Midongy du Sud, in large part overlapping with the Parc National de Befotaka-Midongy du Sud, which is marked on the map. Occurs at an elevation between 950 and 1,600 m.

Redescription of male genitalia (Fig. 5C)

Uncus short, subtriangular, at base with lateral indistinct protrusions dorsally covered with short erect setae directed outwards; apex ventrally provided with bulbous protrusion, terminating with claw-like hook incurved ventrally; valva short, suboval, dully terminated, without extended costal portion; costa and sacculus convex, in terminal portion costa with tiny shallow concavity; outer margin covered with several prominent, erect setae, distinctly longer than those on uncus; vesica membranous with numerous diverticuli and elongate, narrow, tubular ductus ejaculatorius; cornuti in form of multidimensional sclerotised block-like structure in median portion and single elongate sublateral sclerotisation originating close to vesica base and terminating in its distal third.

Contrary to the original description (Griveaud, 1964: 56, Figs. 124–126) the major differences observed in two examined specimens can be summarized as follows (Fig. 5C): uncus not laterally flattened (uncus aplati latéralement), but rather three-dimensional due to ventral bulbous protrusion; valva without elongate costal portion as can be seen in Griveaud’s Figs. 124–125, but rather subquadrate; sclerotised cornuti much more complicated and of different shape compared to Fig. 126 where only single cornutus is visible.

Remarks

Tsarafidynia perpusilla with its red and black colouration is one of the most distinctive Malagasy Syntomini, hard to confuse with any other species. However, the original description of genitalia (Griveaud, 1964: 56, Figs 124–126) is schematic and these illustrations do little justice to their real appearance. Thus, here we redescribe the male genitalia basing on two specimens to make sure that observed differences are not a result of intraspecific variation.

The species has been described by Mabille (1880) as “Aglaope ? perpusilla”, doubtfully placed in the Zygaenidae genus Aglaope Latreille, 1809. In the original description given in Latin the hindwing is divided into two colour zones, but both of them are described as “black” with the use of the same word, what does not tell them apart and is most probably a typo (Alae posticae margine antico usque ad medium alae nigro; caetera pars nigra est, fimbriaque nigra. Alae subtus similes. Corpus nigrum; antennae simplices, nigrae). Currently on the pin of the presumed holotype in NHMUK (NHMUK010354697), of about the right dimensions (about 15.5 mm apex–apex, 16 mm maximum), there is a French-language handwritten label in the writing style of Ch. Oberthür notifying this fact: “Not in accordance with the description. Hindwings are indicated black in the description.” (Pas conforme á la description. Les ailes inférieures sont indiquées noires dans la description). The labels “Madag” and “Aglaope perpusilla Mab.” also attached to this specimen are in a script consistent for P. Mabille. There is no part of the description in French, except that Mabille writes: “♂, 17 mill…Un mâle (coll. H.-G. Smith). Concinna species, sedis incertae”. Jordan (1928) was the first who mentioned this issue in a publication and proposed that the outer hindwing zone should had been referred as “vitreous” (pars vitrea), but he did not see the holotype and guessed that it had gone missing. This may rise from the fact that the type specimens of many species described by P. Mabille have been unrecognized for a long time, as only rarely being directly labelled by him. Viette & Fletcher (1968) finally localised what they considered to be the holotype of A. perpusilla in NHMUK. Rothschild (1911) independently described the species as Micronaclia bicolor based on one female (holotype) and two males collected in Antananarivo by Chulliat. This locality, long devoid of native forest, apparently was not confirmed by Griveaud (1964), and is mentioned separately and in quotation mark. Conspecifity of Aglaope perpusilla and Micronaclia bicolor was, according to Viette (1965), established first in the collection of MNHN by H. de Toulgoët, and then published by Griveaud (1964), who created a separate genus Tsarafidynia for it. Therein, the outer zone of the hindwing should have been referred as red or carmine (certainly not black nor vitreous!) in the original description. The type species of Aglaope, Sphinx infausta Linnaeus, 1767 (Zygaenidae—see also Viette, 1965) which has pectinate antennae and is patterned just like the presumed type of Aglaope perpusilla, black with the basal part of hindwing red. It is likely for this reason alone that Mabille’s description was simply inaccurate, while he wrote the identity label correctly; more likely he meant to write caetera pars rubra est, and the second use of black in the same sentence in any case makes no logical sense.

Holotypes of Tsarafidynia perpusilla and Micronaclia bicolor are deposited in NHMUK London. There are further 15 specimens in MNHN and 35 in PBZT. In these collections are also specimens collected in 1970s, so a few years after the monograph of Griveaud (1964) and range of the species needs to be reassessed including all the specimens.

Discussion

Significance of Réserve Spéciale d’Ambohitantely as a local centre of diversity of Malagasy Syntomini and in protection of the biodiversity of Madagascar

Madagascar is one of the world’s richest biodiversity hotspots (Ganzhorn et al., 2001) with very high levels of endemism (Goodman & Benstead, 2005), attributed to long lasting isolation from other continents and “explosive” evolutionary radiations (Dewar & Richard, 2007; Yoder & Nowak, 2006). An illuminating example of such a radiation within Lepidoptera is the endemic Malagasy lineage of the tribe Syntomini (Przybyłowicz et al., 2019). In the Réserve Spéciale d’Ambohitantely eight species in four genera have been recorded, and five of them are known so far only from this place. This means that 5% of total species-level diversity (Viette, 1990) of the group is currently recorded just from less than two thousand hectares of forest, making the Reserve a centre of the local diversity of Syntomini.

Our results provide further evidence for importance of this Reserve, comprising one of the last considerable fragments of forest in the entire Central Plateau of Madagascar (Ratsirarson et al., 2003), as an important complement for preservation of the remaining biodiversity of Madagascar. The uniqueness of the area is so far underscored by three endemic plants and three endemic frog species (Goodman, Raherilalao & Wohlhauser, 2018). Also, in terms of phylogeography the reserve conserves unique genetic diversity in otherwise widespread species of butterflies (Linares et al., 2009). At the same time, the biodiversity of this place is still not fully documented, especially regarding arthropods. Here we recorded two Syntomini species new for Ambohitantely. In the last few years, several taxa new to science have been described from the Reserve: four species of subsocial Anelosimus spiders (Agnarsson et al., 2015), the mite Atropacarus distinctus Niedbała & Starý, 2014, and two rove beetles: Squamiger elegans Hlaváč & Baňař, 2016 and Ambohitantella banari Hlaváč & Nakládal, 2016, with a new genus created for the latter species.

The montane moist evergreen forest present in Ambohitantely, as well as its floristic and faunistic species composition indicate close affinities with the region of eastern Madagascar (Langrand, 2003; Gautier et al., 2018). Syntomini of the Reserve also match this pattern, as the three more widely distributed species, Maculonaclia ankasoka, Tritonaclia stephania and Tsarafidynia perpusilla, are known from several localities in the eastern and central Madagascar (Griveaud, 1964), and Ambohitantely is one of their northern- and westernmost sites (Fig. 2); moreover, all these localities share the same general type of forest (Gautier et al., 2018). This is also the case in another arctiine genus, Cyana in the Lithosiini (Karisch, 2013; Volynkin, 2020). As the highest diversity of Syntomini species occurs in the longitudinal zone of tropical forests extending throughout the eastern part of the island (Lees, Kremen & Andriamampianina, 1999), it is possible that the entire Syntomini fauna present in the area of Ambohitantely derives from the so called “eastern” forests.

Knowledge on the distribution of Malagasy insects, including Lepidoptera, is still far incomplete, selective and biased towards protected and easily accessible areas (Iannella, D’Alessandro & Biondi, 2019). Malaise trap studies have started to show a remarkable level of previously unknown diversity, even in one of the best studied reserves, Parc National d’Analamazoatra [= “Andasibe”], notably among the micromoths (Lopez-Vaamonde et al., 2019). However, considering all published records (Griveaud, 1964, 1966, 1969, 1970, 1972, 1974; Viette, 1987), we can infer that that the distribution of Syntomini in the area east from Ambohitantely has been studied far more intensively than other regions of the island. An exception for areas readily accessible from the capital is Paysages Harmonieux Protégé du Complexe Anjozorobe-Angavo (“Anjozorobe”), where the fauna of syntomines is still poorly studied (D.C. Lees, 2004–2018, personal observations). This still forested part of the Angavo Massif is the first major patch of forest encountered eastwards of Ambohitantely (ca. 90 km E as the crow flies). Therefore, in terms of understanding the past forest connectedness of Ambohitantely, more intense efforts should be made to examine this area. For example, Tritonaclia stephania was already recorded at a similar elevation along the once unfragmented Angavo Massif, at La Mandraka (Griveaud, 1964). It may turn out that ranges of at least part of the Syntomini species known only from Ambohitantely are wider than currently known, and they are not actually endemic to the area. Moreover, it is supposed that forests of Ambohitantely and Anjozorobe-Angavo were connected to each other only a few hundred years ago (Rakotondravony & Goodman, 1998 in Goodman & Raherilalao, 2003), but there is no direct evidence when the separation occurred, and it was most probably before 1900 (see Linares et al., 2009), if not long before. Rather surprisingly in this context, results of Linares et al. (2009) suggest that Ambohitantely has remained as an isolated patch of forest long enough for genetic drift to fix a unique COI haplotype in three species of Heteropsis butterflies occurring in the Reserve. However, the also dense forest restricted riodinid Saribia tepahi (Boisduval, 1833) exhibited a similar haplotype to the population in Tsaratanana (Linares et al., 2009), 450 km to the north, perhaps suggesting that a forest connection northwards may have existed within the time of human colonisation.

The current landscape of the Central Plateau and its most recent historical forest cover, has been widely debated since the 19th century. Ambohitantely, as one of the last remnants of forest in the Central Plateau, is of particular interest for such speculations. Three main approaches to the vegetational history of the Central Highlands can be distinguished, as summarized in Yoder et al. (2016): the “forest”, “grassland” and “mosaic” hypotheses. Recent studies, not only botanical, but also those on distribution of mouse lemur species, support the latter one. This assumes that landscape composed of patches of forest and grassland had existed in the High Plateau long before human arrival, in cycles of isolation and reconnection of forest fragments driven by climate, and with a rapid fragmentation and separation from the eastern rainforests near the last glacial maximum (Yoder et al., 2016 and papers cited therein; Joseph & Seymour, 2020; Tiley et al., 2020). In that light, referring to the forest fragments of Ambohitantely as “relict” is not quite correct if it presumes Ambohitantely was part of an extensive and continuous former dense tropical forest (Klein, 2004). Despite this, the “lost paradise” line of thinking, connected with the “forest” hypothesis promoted by the French colonizers at the end of 19th century (but see Grandidier, 1898) is still present, influencing discussions on the environmental policy on Madagascar (Pollini, 2010; Amelot, 2017).

However, it is undeniable that current deforestation and fragmentation of the forest is primarily anthropogenic (S.M. Goodman, 2021, unpublished data). The danger to Ambohitantely’s main forest block of fires started in the adjacent grasslands has been at least partly mitigated by installed firebreaks (Goodman, Raherilalao & Wohlhauser, 2018). However, direct deforestation, the second main threat for Ambohitantely, is still increasing at alarming rates, and concerns also forest within the borders of the Reserve. Between 2010 and 2017 alone, over 400 ha of forest disappeared in the Reserve and adjacent areas, whilst the rate of natural forest regeneration since 1949 up to 2017 was virtually negligible. This has caused not only decrease of forest coverage, but also its further fragmentation (S.M. Goodman, 2021, unpublished data).

Effects of this fragmentation in Ambohitantely have already been studied on herpetofauna (Vallan, 2000), birds (Langrand & Wilmé, 1997) and insectivorous mammals (Goodman & Rakotondravony, 2000), but to our knowledge it has never been assessed in regard to invertebrates (except in the context of phylogeography of butterflies: Linares et al., 2009). According to our observations, Malagasy syntomines are rather sedentary species, and do not fly between forest fragments, and thus their relatively abundant fauna can make Ambohitantely an excellent site for the future studies on the mobility of species belonging to this group, as well as potential impact of forest fragmentation on their populations. An especially suitable model could be Thyrosticta dilata with its behaviour, rather unusual among Malagasy Syntomini, of being lured to light in large number of individuals. During our study, 40 of total 45 specimens were attracted in this way. Moreover, all of them were trapped within the forest, and no individual was recorded in the shrubby vegetation of the ecotone (Fig. 1D) nor collected by a light trap set in the grassland, ca. 100 m from the forest’s edge (Fig. 1B).

As already mentioned, further investigation of the distribution of Syntomini in remnant patches of forest in the area of Central Plateau is crucial to understanding their overall patterns of distribution. Also research on biology of Madagascan Syntomini and their potential food resources may shed some light on their ecological connections with certain types of vegetation. At the same time, with five Syntomini species known so far only from Ambohitantely, it cannot be excluded that some of them will turn out indeed to be endemic for the area, as local endemism is characteristic for many evolutionary lineages within the fauna of Madagascar (Wilmé, Goodman & Ganzhorn, 2006), also within arthropods, spectacularly so for examples among the giant pill millipedes (Wesener, 2009), dung beetles (Knopp et al., 2011) or mayflies (Benstead et al., 2003). Up until now several hypotheses have been proposed to explain these unique patterns of distribution, but they were addressed mostly to vertebrates and results show that in many cases a pluralistic approach is required rather than emphasis on a single environmental factor (Wilmé, Goodman & Ganzhorn, 2006; Pearson & Raxworthy, 2009; Vences et al., 2009).

Polymorphism of males of Thyrosticta dilata

We described here a yellow morphotype of Thyrosticta dilata, omitted by Griveaud (1964) in the original description. This species possesses two discrete forms with continuous variation between individuals within each of them. A similar example from the tribe Arctiini is recently described in the Amazonian species Watsonidia fulgida Grados, 2019, where both males and females represent two separate morphotypes within one species, but with continuous variation in male genitalia among specimens of both morphotypes. As for W. fulgida (Grados, 2019), the intraspecific variation in Thyrosticta dilata is not related to any sexual dimorphism (indeed the female remains unknown), nor to geographic, environmental or seasonal dimorphism, because specimens of both types were collected simultaneously in the same place, and moreover the species is known only from the Ambohitantely Reserve. Other, but more phylogenetically distant examples in Lepidoptera are: the Asian clearwing moth Bembecia rushana Gorbunov, 1992 with two differently coloured morphotypes and the African nymphalid Euphaedra eberti Aurivillius, 1896, with two significantly different wing patterns. In both cases, genetic and morphological analyses confirmed conspecifity of the forms (Zúbrik et al., 2019; Garrevoet, Bartsch & Lingenhöle, 2013). However, causes of variation in all three abovementioned species remain unknown (Grados, 2019; Zúbrik et al., 2019; Garrevoet, Bartsch & Lingenhöle, 2013).

As common in Arctiinae moths (Simmons, 2009), nearly all members of Malagasy Syntomini, including Thyrosticta dilata, are aposematically coloured. They usually have black, brown or ochraceous wings as background with white, yellow, orangish-ochraceous and hyaline spots, up to nearly transparent wings, a few species loosely resembling wasps, and often possess black-yellow abdomens (Griveaud, 1964). However, it is not clear that the majority are close wasp mimics, and some species exhibit a possibly Müllerian resemblance with procridine zygaenids that fly in the vicinity (D.C. Lees, Masoala 1993, Ranomafana 1995, Ankazomivady 2003, personal observations). According to the theory of aposematism, each individual should strictly replicate a single pattern that predators learn to avoid. However, numerous examples of variation within species or even single populations are observed across virtually all groups of warning coloured animals (see review by Briolat et al., 2019). Such variation is not uncommon also in tiger moths, and in some cases having an extreme form causes taxonomic complications due to assignation of conspecific males and females to different species (Moraes et al., 2016). This even occurs for Syntomini, e.g., in the genus Pseudothyretes (Przybyłowicz & Tarcz, 2015). According to our observations, other examples of the intraspecific variability among Malagasy Syntomini are shown by some members of genera Stictonaclia and Dubianaclia, and by Thyrosticta cowani Griveaud, 1964, but the taxonomic status of these forms still demands revision. Recently, we described continuous variability in the wing pattern in the Mauritian endemic Dysauxes florida de Joannis, 1906, a species which is shown to derive from the Malagasy radiation (Ł. Przybyłowicz et al., 2021, unpublished data). Numerous and elaborate explanations of the variation within aposematic species have been proposed, but theoretical models often do not meet with observations (Briolat et al., 2019). Variability of colouration in tiger moths can be determined genetically, as for the three major phenotypes of Euplagia quadripunctaria (Poda, 1761) (Liebert & Brakefield, 1990), but causes of this phenomenon in Thyrosticta dilata remain unknown and need further research, especially in the context of similarity of the yellow morphotype to Thyrosticta vieui Griveaud, 1964, which may prove illuminating.

Conclusions

Our results contribute towards an adequate description of diversity of Malagasy Syntomini, indicating that the Réserve Spéciale d’Ambohitantely is a centre of local richness of the group, about 63% of which appear also to be endemic there. It provides further evidence for the importance of the area in protection of the remaining biodiversity of Madagascar. It also highlights the Malagasy Syntomini, whose early stage biology is as yet completely unknown, as an important new study system for the study of adaptive radiation in relation to the diversification of colour pattern.

Supplemental Information

Supplemental Information 1 Barcode sequences used in the paper.

Click here for additional data file.

Supplemental Information 2 Accession numbers of the specimens in the ISEA PAS collection.

Specimens of the yellow morphotype of Thyrosticta dilata bold.

Click here for additional data file.

Supplemental Information 3 Genetic distance (p-distance) between barcode sequences from individuals of Thyrosticta dilata and Maculonaclia tampoketsya with GenBank accession codes.

Codes of the specimens of the yellow morphotype of T. dilata bold. GenBank accession code for the barcode sequence of Fletcherinia decaryi: MK158546.

Click here for additional data file.

We are grateful to Niklas Wahlberg for help with molecular studies, Steven Goodman for providing data from his manuscript in revision and a shapefile with the borders of protected areas of Madagascar, Anna Przystałkowska for graphic processing of figures, Michał Grzyska for consultations of French-language texts, and Joël Minet (MNHN) for a query about a publication date and on previous occasions facilitating access to the MNHN collection (likewise Alberto Zilli and Geoff Martin for access to the NHMUK collection). We thank to MICET (Antananarivo) staff, especially Tiana Vololontiana and Benjamin Andriamihaja for their excellent logistic support and for arranging our permits, students Jimmy Rakotonirina and Hakimou Mahamoudou for help in collecting specimens, Vincent Razafindranaivo (University of Antananarivo) for efficiently arranging the University component of our permit conditions and Balsama Rajemison (PBZT/CAS) and her collection supervisor for access to the Lepidoptera collection and Brian Fisher and the local staff of CAS for helping with logistics of equipment. We thank our drivers and rangers working at the Réserve Spéciale d’Ambohitantely for very reliably facilitating the fieldwork and alerting us about the forthcoming global pandemic before the closure of the Madagascar border just days after the end of this fieldwork. Finally we are grateful to three reviewers for valuable comments on the paper.

Abbreviations

ISEA PAS Institute of Systematics and Evolution of Animals Polish Academy of Sciences, Kraków, Poland

MNHN Muséum national d’Histoire naturelle, Paris, France

NHMUK Natural History Museum, London, United Kingdom

PBZT Parc Botanique et Zoologique de Tsimbazaza, Antananarivo, Madagascar

DCL David C. Lees

ŁP Łukasz Przybyłowicz

Wing Venation

1A + 2A anal vein

CuA1–CuA2 cubital veins

DC discal cell

M1–M3 medial veins

R1–R5 radial veins

Additional Information and Declarations

Competing Interests

Author Contributions

Field Study Permissions

DNA Deposition

Data Availability

The authors declare that they have no competing interests.

Marcin Wiorek conceived and designed the experiments, performed the experiments, analyzed the data, prepared figures and/or tables, authored or reviewed drafts of the paper, and approved the final draft.

Kamila Malik analyzed the data, prepared figures and/or tables, authored or reviewed drafts of the paper, and approved the final draft.

David Lees conceived and designed the experiments, performed the experiments, analyzed the data, authored or reviewed drafts of the paper, and approved the final draft.

Łukasz Przybyłowicz conceived and designed the experiments, performed the experiments, analyzed the data, authored or reviewed drafts of the paper, and approved the final draft.

The following information was supplied relating to field study approvals (i.e., approving body and any reference numbers):

Field collecting was approved by Direction Generale de l’Environment et des Forets and Direction de la Gestion des Ressources Naturelles Renouvelables et des Ecosystemes of the Republic of Madagascar. Field collection was undertaken under permit Nos. 251/06/MINENV.EF/SG/DGEF/DPB/SCBLF/RECH and 292/19/MEDD/SG/DGEF/DGRNE.

The following information was supplied regarding the deposition of DNA sequences:

The cytochrome c oxidase subunit I sequences are available at GenBank: MK158546 and MW817635 to MW817665.

The following information was supplied regarding data availability:

The ISEA PAS collection accession numbers of the specimens are available in the Supplemental Table.

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
