# Peer review of "Malagasy Polka Dot Moths (Noctuoidea: Erebidae: Arctiinae: Syntomini) of Ambohitantely—endemism in the most important relict of Central Plateau rainforest in Madagascar"

_PeerJ, doi:10.7717/peerj.11688_

## Round 0.1 · original submission · Minor Revisions

This is an excellent manuscript that is well written and nicely illustrated. Three reviewers felt that the paper was generally acceptable for publication with minor revisions. I encourage you to carefully consider their comments when revising the paper. I have listed a few additional revisions/edits/critiques below:

1) I would like to suggest that all taxonomic citations in the text be properly cited in the references section.

2) Line 36. Reword “we present the diversity of the tribe Synotomini…”

3) Line 48. Delete “uniquely”

4) Contra one of the reviewers, I believe that the molecular protocols are sufficient as written. I understand this critique but believe that these methods are standard practice and details are unnecessary – I would suggest that the PCR volumes, for example, be deleted.

5) The first section of the Discussion could probably use some reduction in length. It’s more of a long unrelated narrative (mostly) than information relevant to this study. As such it seems a bit disjunct from the main taxonomic focus of the paper.

·

Basic reporting

The reviewed work is an interesting analysis of the endemism of the Réserve Spéciale d’Ambohitantely, on the example of Syntomini. It consists of three parts. The most important part is the redescription and illustration of the adults and genitalia of studied species. The descriptions for the first time of the females of Thyrostictavestigii Griveaud, 1964 and Maculonacliatampoketsya Griveaud, 1969, are especially valuable.
The second part is an analysis of endemism and protection of the studied reserve.
The third part is a description of the polymorphism of males of T. dilata and a general discussion on this topic.
The Authors provide an annotated checklist of the eight species occurring in the Reserve. Two species wider distributed in Madagascar are recorded from the Reserve for the first time. The occurrence of five endemics species described by Griveaud, known only from Ambohitantely was confirmed.
I am a bit surprised that as a result of such intensive field research and analysis of museum material, no new species for science was discovered or no species described by Griveaud was synonymized. The confirmation of 5 endemic species typical for this area, already described by Griveaud, shows that the species composition of Syntonimi of the Reserve is rather stable and Griveaud's systematic research was effective. On the other hand, it indicate that the populations of these species are still preserved and in relatively good condition. This is one of the arguments to protect this area.
Of course, there is no doubt that the studied area is extremely valuable. The concentration of endemic species may also result from the destruction of the remaining areas and the intensity of taxonomic studies in this area.

Experimental design

Methods are described in detail.
In my opinion, the distinction of the collected material captured at night and during the day is unnecessary.

Validity of the findings

I believe that the main goal of these studies was the description or redescription of adults and genitalia of selected species. Without a doubt, it is definitely a scientific achievement of this paper, but I suggest to emphasize the most important diagnostic features, both in terms of external structure and genitals. For instance, I wonder if everted vesica of aegeagus have significantly affect the determination of the species?

Molecular studies have been done and it is valuable, here are used in the study of yellow and black morphotype of T.dilata and female of M.tampoketsya. I suppose the results will be also used in further phylogenetic and systematic analyzes.

Additional comments

The work is written correctly. In my opinion, sometimes is too lengthy and too detailed. Such as, for example, the analysis and remarks on the museum material. On the other hand, it proves a very extensive knowledge of the material and topic.
Sometimes other issues are omitted such as in Methods, the total time of fieldwork in December 2006, 2019, and March 2020. I suggest giving a full range of field research.
The Authors mention that the group greatly requires modern revision, so I suggest adding the photo of M. altitudina in Fig. 3
The description of yellow morphotyp is very detailed. The discussion on polymorphism is valuable, but the source of this phenomenon remains unclear. The mentioned species of Bembecia from Sesiidae are also a good example of polymorphism.
The authors mention the lack of knowledge about the bionomy of these species and their host plants. Perhaps endemism is also a result of interactions with potential endemic host plants. I think it should be the next step in research.
Overall, these are valuable studies of endemism based on the example of Syntomini. I fully recommend the contribution for publication. I have made some corrections/suggestions directly into the manuscript.

Reviewer 2 ·

Basic reporting

No comment

Experimental design

No comment

Validity of the findings

No comment

Additional comments

In this study, the authors provide a checklist for Syntomini occurring within the Réserve Spéciale d’Ambohitantely. Representing an important step in describing the biodiversity of Madagascar, which is of crucial importance due to the biodiversity crisis. I believe this to be of interest to a wide audience.
Overall the manuscript if well written and easy to read.
I have two minor suggestions:
1. To make numbers, such as elevations easier to read you could present in as 1,250 for example

2. LINE 198: some more details regarding the PCR would be helpful. For example including the concentration of the primers used, and PCR conditions and what sequencer was used for the sequencing.
For example this could read something along these lines:
“PCR was carried out in a 10 µl volume, which included #µl of each Primer (10µM) and the use of hot-start ready PCR mix (StartWarm HS-PCR Mix, A&A Biotechnology, Poland). Reactions were performed under the following conditions…. and DNA sequencing performed on a … Sequencer.”

·

Basic reporting

The authors of the manuscript investigated fauna of Syntomini moths in Ambohitantely, Madagascar. This place represents one of the last forest fragment on Madagascan Central Plateau. These moths belong to a group with incredible high amount of species endemic to Madagascar (99) and five species are known from this area only. The work contains all relevant references. The structure follows classical paper structure. The paper is not experimental and thus does not contain any hypothesis.

Experimental design

The authors visited the research place three times in years 2006, 2019 and 2020, collected the material using light traps and also by a focal sampling during day. Additionally, the authors also investigated material from four national museum collections. As a minor part, the authors also sequenced several species and conducted comparison of their genetic information using classic barcoding and usual software. However, the authors should explain in more detail the legend to Fig. 6, i.e. what means the numbers on the tree (bootstrap values?) and the scale bar.

Validity of the findings

During the survey, the authors documented 7 species of target moth group (58 specimens in total), they failed to find one additional species for which exists a published record. Only three of these recorded species are known to have a wider distribution across Madagascar island.
I see the findings interesting and as the authors are known experts on Madagascan Lepidoptera, there is no doubt about validity of their findings. The study is well written, the specimens are as good as they are and evidently the figures were not manipulated. The figured male and female genitalia of the moths are important for Id of the species by other experts.
So my only concerns is whether such a rather local study is of interest of a wider audience of readers. On the other hand, the fauna of Madagascar is seriously threatened and especially the last fragments of forest of the central Madagascan highland. Thus leave a final decision to Editor.

Additional comments

I have no additional comments

---

## Round 0.2 · accepted · Accept

I really appreciate the careful attention to the reviewers' comments. You did a great job revising the paper and reducing the overall length of the discussion. Great contribution to Peerj and moth literature/taxonomy!